# Recovering Cobalt and Sulfur in Low Grade Cobalt-Bearing V–Ti Magnetite Tailings Using Flotation Process

**Junhui Xiao** [1,2,3,4,*] and **Yushu Zhang** [5]

1   School of Environment and Resources, Southwest University of Science and Technology, Mianyang 621010, China
2   Key Laboratory of Sichuan Province for Comprehensive Utilization of Vanadium and Titanium Resources (Panzhihua University), Panzhihua 61700, China
3   Sichuan Engineering Laboratory of Non-Metallic Mineral Powder Modification and High-Value Utilization, Southwest University of Science and Technology, Mianyang 621010, China
4   Key Laboratory of Ministry of Education for Solid Waste Treatment and Resource Recycle, Southwest University of Science and Technology, Mianyang 621010, China
5   Institute of Multipurpose Utilization of Mineral Resources, Chinese Academy of Geological Sciences, Chengdu 610041, China
*   Correspondence: xiaojunhui33@163.com; Tel.: +86-139-9019-0544

**Abstract:** There is 0.032% cobalt and 0.56% sulfur in the cobalt-bearing V–Ti tailings in the Panxi Region, with the metal sulfide minerals mainly including $FeS_2$, $Fe_{1-x}S$, $Co_3S_4$, and $(Fe,Co)S_2$, and the gangue minerals mainly including aluminosilicate minerals. The flotation process was used to recover cobalt and sulfur in the cobalt-bearing V–Ti tailings. The results showed that an optimized cobalt–sulfur concentrate with a cobalt grade of 2.08%, sulfur content of 36.12%, sulfur recovery of 85.79%, and cobalt recovery and 84.77% were obtained by flotation process of one roughing, three sweeping, and three cleaning under roughing conditions, which employed pulp pH of 8, grinding fineness of <0.074 mm occupying 80%, flotation concentration of 30%, and dosages of butyl xanthate, copper sulfate, and pine oil of 100 g/t, 30 g/t, and 20 g/t, respectively. Optimized one sweeping, two sweeping, and three sweeping conditions used a pulp pH of 9, and dosages of butyl xanthate, copper sulfate, and pine oil of 50 g/t, 15 g/t, 10 g/t; 25 g/t, 7.5 g/t, 5 g/t; 20 g/t, 5 g/t, 5 g/t, respectively. Optimized one cleaning, two cleaning, and three cleaning condition dosages of sodium silicate of 200 g/t, 100 g/t, 50 g/t, respectively. Study of analysis and characterization of cobalt–sulfur concentrate by X-ray diffraction (XRD), automatic mineral analyzer (MLA), scanning electron microscopy (SEM), and energy dispersive spectroscopy (EDS) showed that the main minerals in cobalt–sulfur concentrate are $FeS_2$, $Co_3S_4$ and $(Fe,Co)S_2$, of which $FeS_2$ and $(Fe,Co)S_2$ accounted for 65.64% and $Co_3S_4$ for 22.64%. Gangue minerals accounted for 11.72%. The element Co in $(Fe,Co)S_2$ is closely related to pyrite in the form of isomorphism, and the flotability difference between cobalt and pyrite is very small, which makes it difficult to separate cobalt and sulfur. Cobalt–sulfur concentrate can be used as raw material for further separation of cobalt and sulfur in smelting by pyrometallurgical or hydrometallurgical methods.

**Keywords:** cobalt; pyrite; linneite; cobalt pyrite; V–Ti magnetite tailings; flotation

## 1. Introduction

As an important strategic material metal, cobalt plays an important role in national economic development. It is also an important raw material for the production of various high temperature and

corrosion resistant alloys, cemented carbides, superhard materials, magnetic materials, catalysts, and other materials [1]. It is widely used in industries such as aviation, aerospace, electrical appliances, machinery manufacturing, chemical industry and ceramics, etc. Cobalt–sulfur concentrate is the main raw material for extracting cobalt in industry. Pyrite is the main sulfur-bearing mineral in cobalt–sulfur concentrate [2]. As cobalt–sulfur concentrate mainly contains pyrite, pyrrhotite, and gangue minerals such as talc, quartz, and chlorite, it is sometimes difficult to obtain cobalt concentrate and sulfur concentrate separately.

Cobalt-bearing minerals are mainly composed of arsenide, such as cobaltite (arsenopyrite), arsenopyrite, orthoclase arsenopyrite, and skutterudite, etc.; sulfide, such as sulfur–copper–cobalt ore, and sulfur–nickel–cobalt ore, etc.; oxide, such as cobalt oxide, cobalt earth ore, miscellaneous cobalt ore, and spheroidite; and the smelting slags of cobalt-bearing pyrite, cobalt–nickel ore, and cobalt-bearing copper ore [3].

When separating cobalt minerals from cobalt ores and cobalt-bearing ores, the flotation method is mostly employed, and in rare cases, the gravity beneficiation method is also used. Gravity beneficiation method is mainly used for arsenide with larger specific gravity (e.g., cobaltite with a specific gravity of 6.2 and arsenic–cobalt with a specific gravity of 6.5). As there is small difference between the specific gravity of cobalt oxide mineral and that of gangue, it is difficult to separate the cobalt oxide mineral (the specific gravity of cobalt wafer is 3, and the specific gravity of cobalt earth ratio is 3–3.5) from gangue effectively by gravity beneficiation method, thus making it unable to obtain an ideal cobalt concentrate product [4,5].

If cobalt occurs in pyrite, pyrrhotite, chalcopyrite, and other minerals in the form of isomorphism, cobalt concentrate is usually recovered by flotation carrier minerals rather than directly obtained. If cobalt-bearing minerals are cobalt-rich ores and cobalt sulfide ores with good floatability, it is more economical to recover cobalt concentrate by flotation. However, the problem is that cobalt minerals are closely related to other metal minerals and gangue minerals, and their floatability has little difference. Therefore, the emphasis is to develop new collectors of cobalt minerals and effective depressants of gangue minerals [6].

At present, cobalt minerals are generally recovered by flotation process. The commonly used collectors for flotation of cobalt-bearing minerals are black catching agent, xanthate, amines, palm oil, fatty acids, and so on. However, there are still no specific collectors for cobalt-bearing minerals so far. A lot of research work is needed to find a kind of collector with high efficiency and good selectivity, which is not highly influenced by other factors in pulp.

There are some SKARN-TYPE deposits, in which pyrite contains a small amount of cobalt. It is of great significance to comprehensively recover drilling metal from concentrators in these mines. However, some factories can only produce sulfur concentrate because they can't reach the cobalt concentrate target, while others dare not adopt flotation technology for various reasons, which wastes a lot of national resources [7]. Therefore, it is of practical significance to study the flotation of pyrite and cobalt-bearing pyrite. The pyrite shows good floatability. Nevertheless, as the geological conditions and mineral composition of each deposit are different, the flotation process of pyrite is also different.

Tieshanhe iron ore in Jiyuan occurs in the hydrothermal deposit of dolomite related to diorite metasomatism. Gangue minerals are mainly tremolite, actinolite, calcite, dolomite, and chlorite. It is difficult to separate tremolite and actinolite from pyrite, and it is difficult to obtain high quality sulfur–cobalt concentrate by multistage concentration. In order to obtain high quality sulfur–cobalt concentrate, it is necessary to inhibit these magnesium-containing silicate minerals in the flotation process. At present, there is a shortage of cobalt metal in China, and the cobalt grade of sulfur and cobalt concentrate recovered by some concentrators is about 0.30%. Using low-grade sulfer–cobalt concentrates for smelting has resulted in high cost for smelters. Therefore, it is an urgent task to improve the quality of cobalt sulfide concentrate [8].

The Wissokogolsk magnetite contains copper and cobalt, and the Ural sulfide magnetite also contains copper and cobalt. Their comprehensive recovery rates of the magnetite are as follows: The

former has a cobalt concentrate grade of 0.50% and a recovery of 36.82%, while the latter has a cobalt concentrate grade of 0.49% and a recovery of 66.40%. The cobalt grade of Jiyuan iron ore is 0.023% and the sulfur grade is 1.14%. The cobalt grade in sulfur–cobalt concentrate is 0.62%, sulfur grade is 41.37%, and cobalt recovery rate is 65.43% [9].

The Panxi Region of China is rich in mineral resources. The proven reserves of vanadium–titanium magnetite are 10 billion tons. Among them, iron resources account for 20% of the domestic iron ore reserves, vanadium resources account for 62% of the national vanadium reserves, and titanium resources account for 90.5% of the national titanium resources reserves. In addition, there are 900,000 tons of cobalt, 700,000 tons of nickel, 250,000 tons of scandium, and 180,000 tons of gallium, as well as a large amount of copper, sulfur, and other resources. However, the comprehensive utilization rate of nonferrous metal resources in the Panxi Region is very low. In the four major mining areas, only the Taihe mining area is recovered, while the other three mining areas are not utilized [10]. The low-grade and scattered distribution of sulfur and cobalt resources in vanadium–titanium magnetite make it difficult to utilize them directly from the original ore [11,12]. Most of the cobalt associated with vanadium–titanium magnetite occurs in the form of pyrite, pyrrhotite, and cobalt–nickel pyrite, while a very small amount of cobalt occurs in the form of pyrite. Due to the small flotability differences among pyrite, pyrrhotite, cobalt–nickel pyrite, and cobalt sulfide ore, which could basically be characterized by the surface properties of pyrite minerals, it is difficult to recover some independent minerals by flotation.

Most cobalt minerals are associated with other metal minerals. At present, valuable cobalt-minerals mainly include cobaltite, skutterudite, chalcopyrite, sulfur–cobalt ore, nickel–cobalt ore, hydrocobalt ore, erythrite, and ferromanganese-bound cobalt ore, etc. The most effective way to recover cobalt minerals is flotation, but manual and repeated screening are also employed. Flotation is the most important method to treat cobalt-bearing minerals [13,14]. Cobalt concentrate can be obtained directly by flotation of single cobalt minerals such as cobaltite and sulfur–cobalt ore. Cobalt concentrate occurring in pyrite and chalcopyrite cannot be obtained directly by the carrier flotation.

Therefore, recovered cobalt from vanadium ilmenite tailings is essentially a mixture of minerals of pyrite, pyrrhotite, cobalt pyrite, and sulfur–cobalt ore. In this paper, the experimental study on the recovery of cobalt and sulfur from cobalt-bearing vanadium titanomagnetite tailings in the Panxi Region was carried out by a flotation process, providing an important research basis for the associated cobalt and sulfur resources of vanadium titanomagnetite tailings in the Panxi Region.

## 2. Materials and Methods

### 2.1. Sampling

The ore samples used in this study were from low grade cobalt-bearing V–Ti magnetite tailings produced by a concentrator in the Panxi Region after concentrating iron and titanium. The water content of the tailings was less than 1% and the particle size was between 0.038 and 20 mm. It was found that the bulk ore samples were all caused by fine-grained caking. In order to avoid large sampling errors, the agglomerated samples were crushed and then shrunk in advance for reserve. The main chemical composition analysis of the samples is shown in Table 1, the phase analysis of cobalt is shown in Table 2, and the X-ray diffraction pattern of the samples is shown in Figure 1. The mineral composition of the ores is complex, with the mineral surface covered with more sludge and the particles bonded to each other. The metal minerals mainly include pyrite, cobalt pyrite, sulfur–cobalt ore, pyrrhotite, and ilmenite. Gangue minerals mainly include titanopyroxene, plagioclase, serpentine, chlorite, mica, amphibole, olivine, and calcite, etc.

**Table 1.** Main chemical composition analysis of cobalt containing V–Ti magnetite tailings (%).

| Composition | Fe | Co | TiO$_2$ | S | Ni | Na$_2$O | MgO | Al$_2$O$_3$ | CaO | SiO$_2$ |
|---|---|---|---|---|---|---|---|---|---|---|
| Content | 10.38 | 0.032 | 2.12 | 2.68 | 0.006 | 0.68 | 6.28 | 12.40 | 12.56 | 42.72 |

**Table 2.** Cobalt chemical phase analysis of cobalt containing V–Ti magnetite tailings (%).

| Composition | Cobalt Sulphide | Cobalt Oxide | Cobalt in Silicate | Totals |
|---|---|---|---|---|
| Content | 0.0301 | 0.0008 | 0.0001 | 0.031 |
| Distribution | 97.10 | 2.58 | 0.32 | 100.00 |

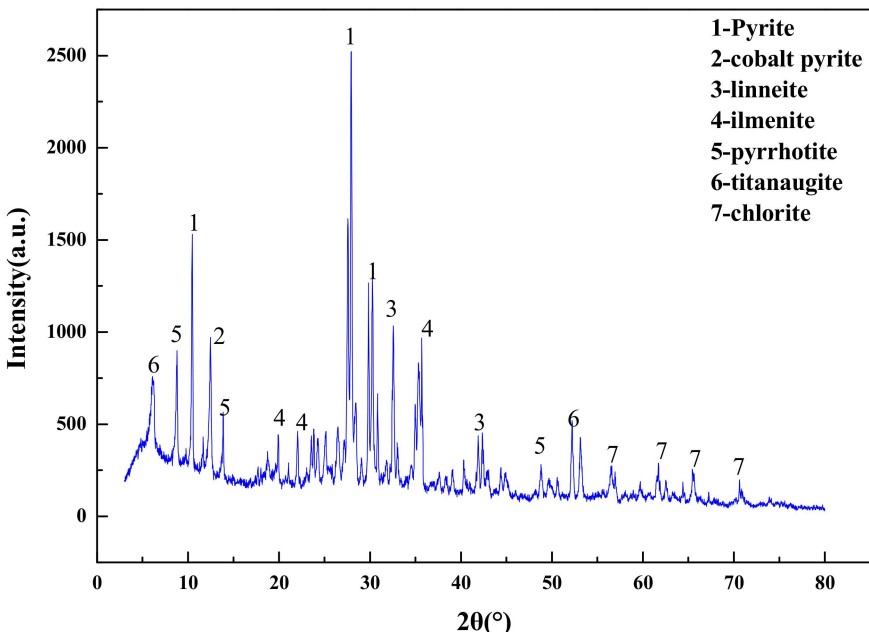

**Figure 1.** XRD analysis of cobalt-bearing V–Ti magnetite tailings.

## 2.2. Chemical Reagent

The main chemical reagents used in this test are ethyl xanthate, butyl xanthate, amyl xanthate, black xanthate, ethyl thionitrogen, copper sulfate, sodium carbonate, and sodium silicate, all of which have analytical purity and a producing area in Tianjin Tianli Chemical Regeant, Co., Ltd., Tianjin China.

## 2.3. Flotation

Flotation (roughing, scavenging), aimed at increasing the grade and recovery of cobalt, was carried out using an XFD-1.5L hanging tank flotation machine (Jinlin Exploration Machinery Plant, China) operating at a spindle speed of 1650 r/min. A 500 g mass of cobalt that contained vanadium–titanium magnetite tailings was added to the 1.5 L flotation tank. Flotation (cleaning), aimed at increasing the cobalt grade, was carried out using an XFD-1.0 L hanging tank flotation machine (Jinlin Exploration Machinery Plant, China) operating at a spindle speed of 1650 r/min. A 300 g mass of cobalt containing cobalt–sulfur concentrate was added to the 1.0 L flotation tank. Distilled water (1.0 L) was added and the pulp stirred and mixed for 3 min, followed by adjustment to the required pH using sodium carbonate. After 10 min of pulping, the regulators (depressant or activator) were added to the slurry and conditioned for 3 min. Then, the collectors for improving cobalt grade and recovery were added and agitated for 3 min. Before aeration, the frothers (pine oil) for improving the bubble were added, with another 3 min of stirring. After 3 min of flotation, the froth (cobalt–sulfur concentrate) and in-tank product (flotation tailings) were separately filtered, dried at 80 °C for 4 h, and weighed. Quantitative analyses of cobalt grade were conducted to calculate the cobalt recovery.

### 2.4. Analysis and Characterization

The chemical composition of solid materials was analyzed by Z-2000 atomic absorption spectrophotometer (Hitachi Co., Ltd.), the diffraction grating was zenier-tana type, 1800 lines /mm, the flash wavelength was 200 nm, the wavelength range was 190~900 nm, the automatic peak seeking setting, and the spectral bandwidth was divided into 4 grades (0.2, 0.4, 1.3, and 2.6 nm) for the analysis of mineral chemical composition.

The phase composition of solid substances (cobalt–sulfur concentrate and cobalt-containing vanadium–titanium magnetite tailings) was analyzed by X-ray diffraction (XRD, X Pert pro, Panaco, The Netherlands).

The microstructure of the solid products was observed by SEM (S440, Leica Cambridge LTD, Germany) equipped with an energy dispersive X-ray spectroscopy (EDS) detector (UItra55, Carl zeissNTS GmbH, Germany).

The chemical phase composition of cobalt sulfur concentrate was analyzed by mineral liberation analyzer (MLA) (FEI electronic optics co., LTD, Australia.), which is composed of Quanta 250 environmental scanning electron microscope, EDAX spectrometer, and jktech-mla3.0 process mineralogy automatic test software. Test conditions: Working voltage 25 kV, magnification 300 times, beam Spot 6.8, particle minus minimum size 30 (pixel).

## 3. Results and Discussion

### 3.1. Cobalt Separation Test

#### 3.1.1. Effect of Different Collectors

The common collectors of sulfide ores are ethyl xanthate, butyl xanthate, amyl xanthate, black catching agent, and ethyl sulfur–nitrogen. Combined with the occurrence of cobalt in tailings of vanadium titanomagnetite and the existing practice and theory of the sulfide flotation process, the comparative tests of ethyl xanthate, butyl xanthate, pentyl xanthate, and black catching agent were carried out to investigate the flotation effect of collectors on cobalt-bearing minerals. Conditions employed were $Na_2CO_3$-adjusted pulp pH of 8, collector dosage of 20 g/t, pine oil dosage of 20 g/t, grinding fineness of <0.074 mm occupying 70%, and flotation concentration of 25%. The results are shown in Figure 2.

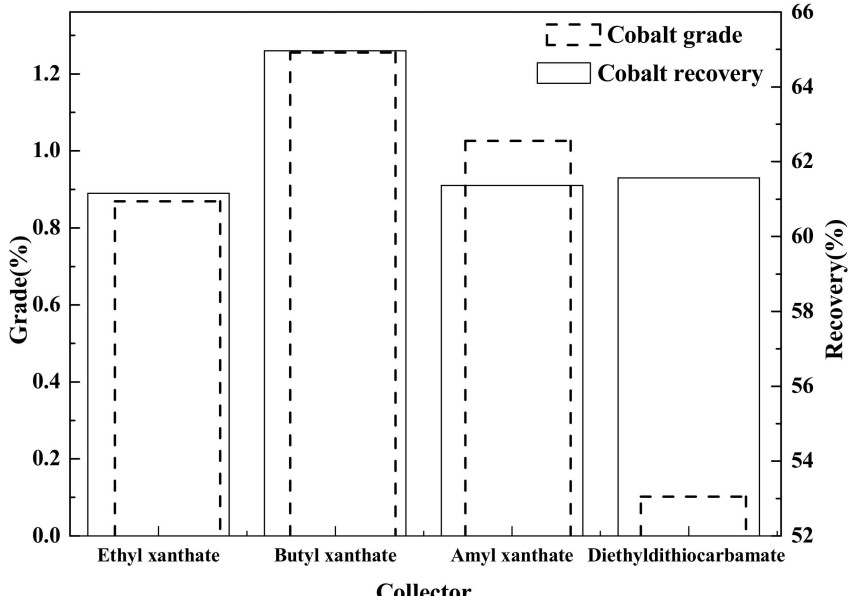

**Figure 2.** Effect of collectors on cobalt grade and recovery in flotation.

Different kinds of collectors have obvious effects on cobalt grade and recovery. Butyl xanthate as a collector has ideal separation index of cobalt, with a cobalt grade of 1.26% and cobalt recovery of 64.92%. The cobalt grades of ethyl xanthate, pentyl xanthate, and black catching agent as flotation collectors have improved slightly compared with that of raw ore, and the cobalt grade in tailings is also higher. Also, this shows that butyl xanthate as collector is more advantageous for cobalt separation, and there are many factors affecting the cobalt separation index. Different collectors have different collectivity and selectivity for different sulfide minerals [15,16]. Therefore, butyl xanthate is a suitable collector for cobalt–sulfur flotation. The separation index of cobalt–sulfur mixed concentrate can be obtained, with a cobalt grade of 1.26% and cobalt recovery of 64.92%.

### 3.1.2. Effect of Collector Dosage

A conical ball mill was applied to grind low-grade cobalt-bearing V–Ti magnetite tailings. A mass of 500 g was processed using a liquid-to-solid ratio R = 2:1; grinding fineness of % of <0.074 mm occupying 70%, flotation concentration of 25%, and pine oil dosage of 20 g/t were prepared to carry out the separation tests with different collectors. The collector dosage of 20 g/t. The results are shown in Figure 3.

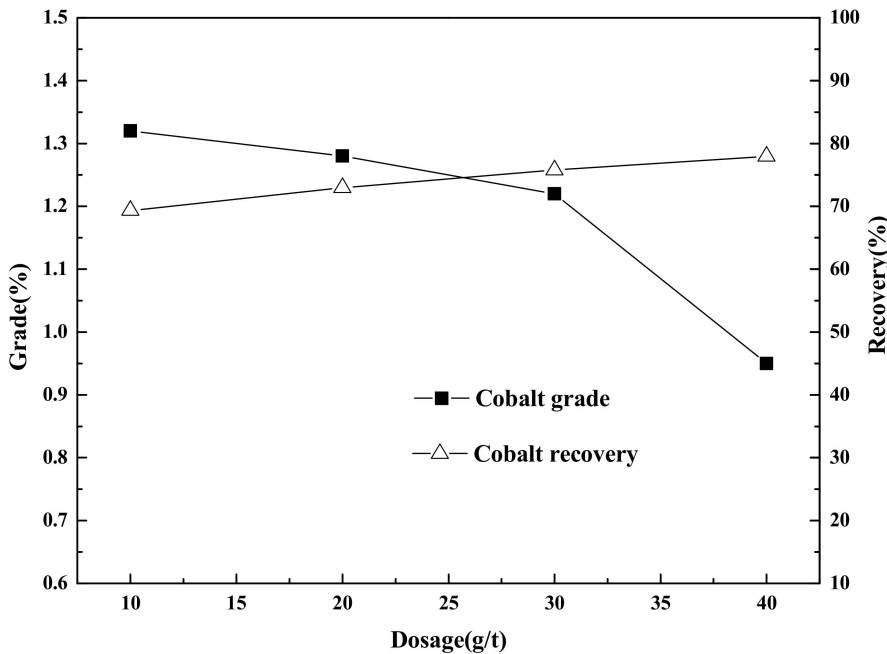

**Figure 3.** Effect of butyl xanthate dosage on cobalt grade and recovery in flotation.

In the flotation process, increasing the amount of collector is beneficial to improve the recovery of concentrate, but the excessive amount of collector will result in the flotation of other nonpurpose minerals, thereby affecting the grade of concentrate [17–19].With the increase of butyl xanthate dosage, the grade of cobalt–sulfur concentrate decreases regularly, and the recovery of cobalt increases accordingly. When the dosage of butyl xanthate increased to 30 g/t, the cobalt grade was 1.22%, and the cobalt recovery rate was 75.78%. Compared with employing a butyl xanthate dosage of 40 g/t, the cobalt grade increased by 0.27%, the cobalt recovery rate decreased by 2.18%, but the tailings difference was 0.01%. Compared with employing 20 g/t of butyl xanthate, the cobalt grade decreased by 0.06% and the cobalt recovery rate increased by 2.84%. This shows that increasing the dosage of butyl xanthate is beneficial to improving the cobalt grade as well as the cobalt recovery of cobalt–sulfur concentrate, but when the dosage is increased to a certain extent, the degree of improvement of the separation index of cobalt–sulfur concentrate is limited. Therefore, a reasonable dosage of collector is not only conducive to improving the flotation index, but also plays a reasonable role in regulating

the cost of reagents. So, the suitable dosage of butyl xanthate is 30 g/t and the separation index of cobalt–sulfur concentrate can be obtained accordingly, with a cobalt grade of 1.22% and cobalt recovery of 75.78%.

### 3.1.3. Effect of Activator Dosage

A conical ball mill was applied to grind low grade cobalt-bearing V–Ti magnetite tailings. As above, a mass of 500 g was processed using a liquid-to-solid ratio R = 2:1; grinding fineness of % of <0.074 mm occupying 70%, flotation concentration of 25%, butyl xanthate dosage of 30 g/t, and pine oil dosage of 20 g/t were prepared to investigated the effects of activator dosage on cobalt grade and recovery. The results are shown in Figure 4.

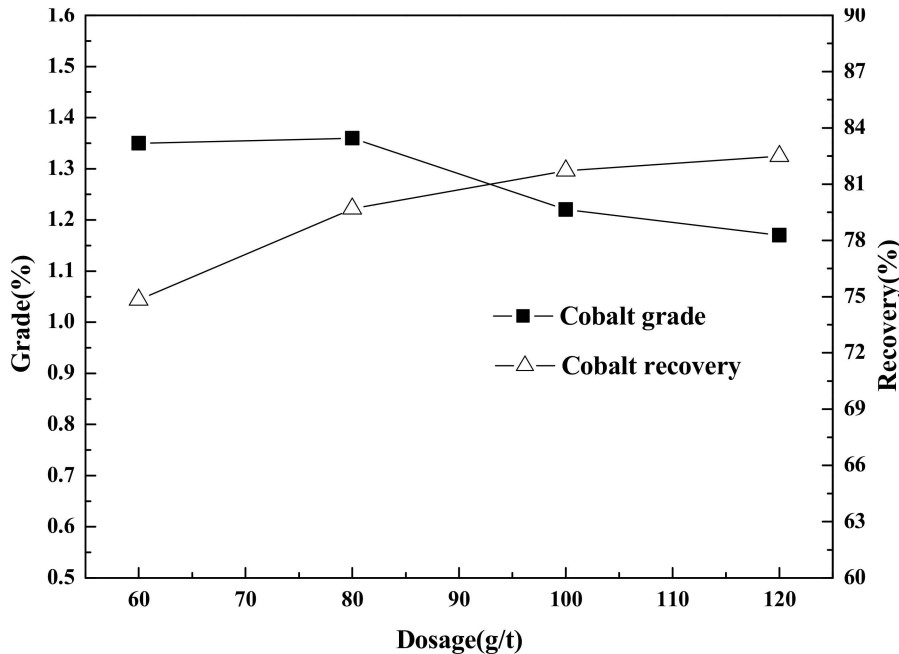

**Figure 4.** Effect of copper sulfate dosage on cobalt grade and recovery in flotation.

The main sulfide minerals in low-grade vanadium titanomagnetite tailings are pyrite, pyrrhotite, cobalt-nickel pyrite, cobalt–sulfur ore, and sulfur–nickel–cobalt ore. Adding appropriate activators in the flotation process is conducive to improving the flotation index. Copper sulfate is commonly used as activator of sulfide minerals such as pyrite, pyrrhotite, and cobalt-nickel pyrite [20,21]. Adding sulfuric acid as activator of the cobalt flotation process has an obvious effect on improving cobalt grade and recovery. $Cu^{2+}$ is utilized to activate sulfide minerals to improve the hydrophobicity of their surface. However, the amount of $Cu^{2+}$ has a more obvious effect on the whole flotation system. The excessive amount of $Cu^{2+}$ has a negative effect on the concentration and separation of the flotation concentrate [22,23]. With the increase of copper sulfate dosage, the cobalt grade decreases regularly, and the cobalt recovery rate increases regularly. Compared with noncopper sulfate, the cobalt grade increased slightly, but the recovery rate of cobalt was higher obviously. However, when the amount of copper sulfate increased to 120 g/t, the cobalt grade decreased to 1.17%. This indicated that the excessive amount of $Cu^{2+}$ would interfere with the separation of cobalt and have a negative impact on the cobalt grade. Therefore, it is important to select a reasonable amount of $Cu^{2+}$ to improve the flotation index. Considering comprehensively, the suitable amount of copper sulfate is 100 g/t, and the separation index of cobalt–sulfur concentrate with a cobalt grade of 1.22% and cobalt recovery of 81.71% can be obtained.

### 3.1.4. Effect of Grinding Fineness

A conical ball mill was applied to grind low-grade cobalt-bearing V–Ti magnetite tailings. As above, a mass of 500 g was processed using a liquid-to-solid ratio R = 2:1. The effect of grinding fineness on cobalt grade and recovery tests were carried out using $Na_2CO_3$-adjusted pulp pH of 8, butyl xanthate dosage of 30 g/t, copper sulfate dosage of 100 g/t, pine oil dosage of 20 g/t, and flotation concentration of 25%. The results are shown in Figure 5.

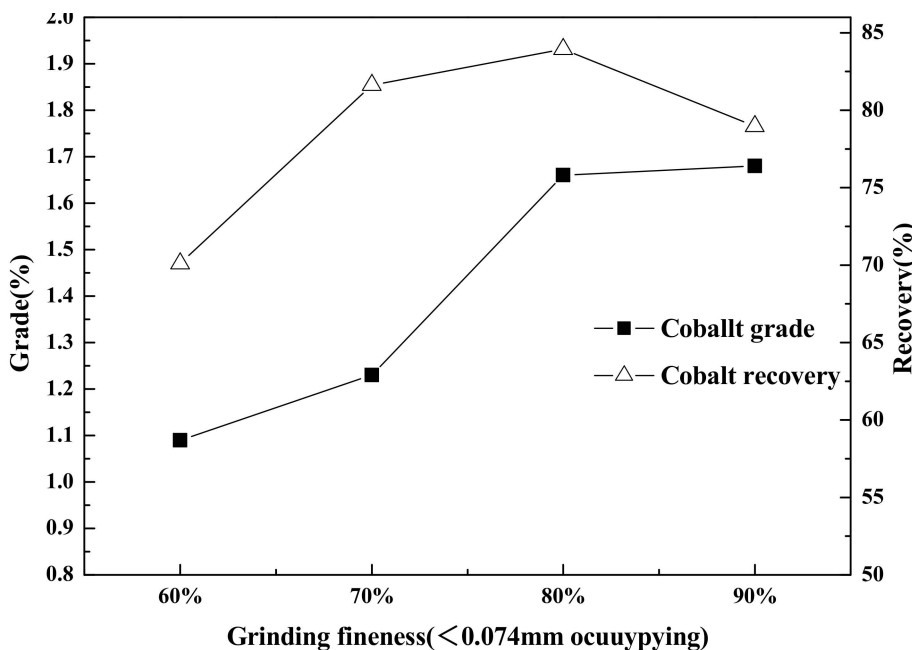

**Figure 5.** Effect of grinding fineness on cobalt grade and recovery in flotation.

The basic dissociation of minerals is the primary factor for realizing mineral separation. In the process of ore separation, grinding fineness is a fundamental parameter of reaction mineral dissociation. With the increase of grinding fineness, the dissociation degree of mineral monomer increases correspondingly, which has a positive effect on flotation reagent and mineral surface. However, over-fine particle size, increased specific surface area of particles, increased consumption of flotation reagent per unit, and high grinding fineness can easily lead to over-grinding and increase grinding cost [24,25]. On the contrary, the reduction of grinding fineness and the degree of mineral dissociation is not conducive to the role of flotation reagents and mineral surface, and will also have a negative impact on flotation indicators. Greater grinding fineness improved cobalt grade regularly, and the recovery of cobalt increased first and then decreased accordingly. When the grinding fineness was increased to <0.074 mm occupying 80%, the cobalt grade was increased to 1.66% and the recovery rate was increased to 83.94%. Compared with the grinding fineness of <0.074 mm accounting for 90%, the cobalt grade was 0.02% lower and the cobalt recovery rate increased by 4.98%. This shows that with the increasing of the grinding fineness and the reduction of the particle size, the interference between mineral particles was intensified, which greatly affects the separation of cobalt. Therefore, the selection of reasonable grinding fineness has an important influence on the flotation separation of cobalt from titanium tailings of vanadium titanomagnetite. So. it is suitable for the grinding fineness of <0.074 mm occupying 80%, which could foster the separation index of cobalt sulfur concentrate, with a cobalt grade of 1.66% and cobalt recovery of 83.94%.

### 3.1.5. Effect of Pulp pH

A conical ball mill was applied to grind low-grade cobalt-bearing V–Ti magnetite tailings. As above, a mass of 500 g was processed using a liquid-to-solid ratio R = 2:1. The effect of pulp pH on

cobalt grade and recovery test were carried out using Na$_2$CO$_3$-adjusted pulp pH of 8, grinding fineness of <0.074 mm occupying 80%, butyl xanthate dosage of 30 g/t, copper sulfate dosage of 100 g/t, pine oil dosage of 20 g/t, and flotation concentration of 25%. The results are shown in Figure 6.

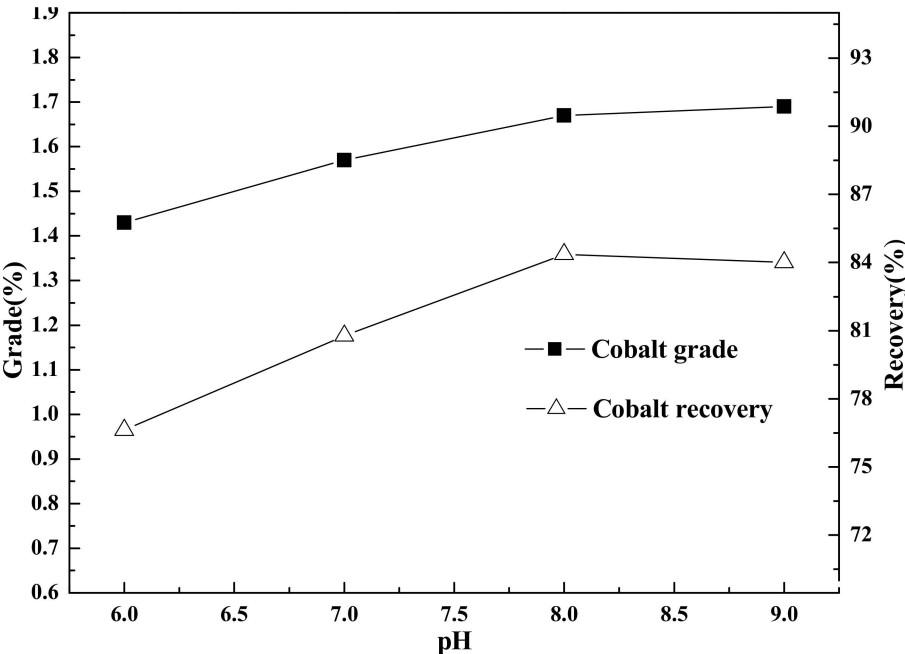

**Figure 6.** Effect of pulp pH on cobalt grade and recovery in flotation.

The pH value of pulp in the flotation process has a great influence on the adsorption of flotation reagent and mineral surface. As a collector of sulfide ore, xanthate generally requires that the pH value of flotation pulp be weak acidic or above. In addition, the pH quality of flotation pulp has a great influence on the treatment of tailings from subsequent beneficiation [24,25]. The cobalt grade and recovery rate increased regularly with the increase of pulp pH value from Figure 6. When the pulp pH value was 9, the cobalt grade of cobalt sulfur concentrate was 1.69%, and when the pulp pH value was 9, the cobalt recovery rate of cobalt sulfur concentrate was 84.37%. Comparing the results when pulp pH = 8 and 9, respectively, the cobalt grade was reduced by 0.02% and the recovery of cobalt was increased by 0.37%. So, it is ideal to realize mineral separation under appropriate pulp pH value. This shows that the pulp pH = 8 is reasonable, which can reduce the consumption of sodium carbonate and foster the separation index of cobalt sulfur concentrate, with a cobalt grade of 1.67% and cobalt recovery of 84.37%.

### 3.1.6. Effect of Flotation Concentration

A conical ball mill was applied to grind low-grade cobalt-bearing V–Ti magnetite tailings. As above, a mass of 500 g was processed using a liquid-to-solid ratio R = 2:1. The effect of flotation concentration on cobalt grade and recovery tests were carried out using Na$_2$CO$_3$-adjusted pulp pH of 8, grinding fineness of <0.074 mm occupying 80%, butyl xanthate dosage of 30 g/t, copper sulfate dosage of 100 g/t, and pine oil dosage of 20 g/t. The results are shown in Figure 7.

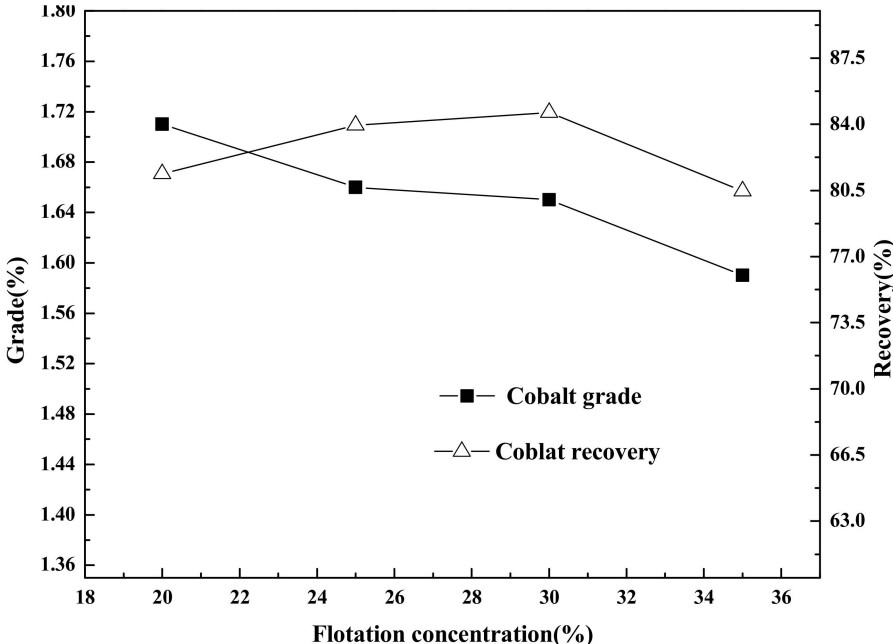

**Figure 7.** Effect of flotation concentration on cobalt grade and recovery in flotation.

Flotation concentration is another important factor affecting the flotation separation index. In the flotation production process, flotation concentration is a parameter of unit capacity of reaction separation equipment. With the increase of flotation concentration, the processing capacity of unit equipment of flotation machines increases; on the contrary, its processing capacity decreases [26–29]. The cobalt grade decreased with the increase of flotation concentration, and the recovery of cobalt increased first and then decreased. This was mainly due to the reduction of flotation concentration and the increase of flotation reagent consumption per unit pulp, which was beneficial to the improvement of cobalt grade of cobalt–sulfur concentrate, but not conducive to the improvement of cobalt recovery rate. The high flotation concentration aggravated the interference between mineral particles, affected the role of minerals and flotation reagents, and was not conducive to improving hydrophobicity of target minerals and hydrophilicity of nontarget minerals. When the flotation concentration was increased to 35%, the cobalt grade of cobalt–sulfur concentrate decreased by (1.65−1.59)% = 0.06%, and the recovery of cobalt decreased by (84.62−80.46)% = 4.16%, compared with the results with the flotation concentration of 30%. Therefore, the reasonable flotation concentration should be 30%, which could foster the separation index of cobalt–sulfur concentrate, with a cobalt grade of 1.65% and cobalt recovery of 84.62%.

### 3.1.7. Effect of Time on Flotation Scavenging Process

Through a flotation process of one roughing, the effect of different flotation conditions on the separation of cobalt was studied in the titanium tailings of vanadium–titanium magnetite. The separation indexes of cobalt–sulfur concentrate with a cobalt grade of 1.65% and cobalt recovery of 84.75% were obtained. Due to fact that the main isomorphic forms of cobalt occur in cobalt-bearing minerals such as pyrrhotite and cobalt–nickel pyrite, it is difficult to obtain an ideal separation index of cobalt–sulfur concentrate by primary flotation. The content of cobalt in flotation tailings is 0.016%. In order to further improve the comprehensive recovery rate of cobalt from cobalt–sulfur concentrate, the effect of different sweeping times on cobalt grade and recovery rate was investigated by increasing sweeping times. The technological process is shown in Figure 8 and the results are shown in Table 3.

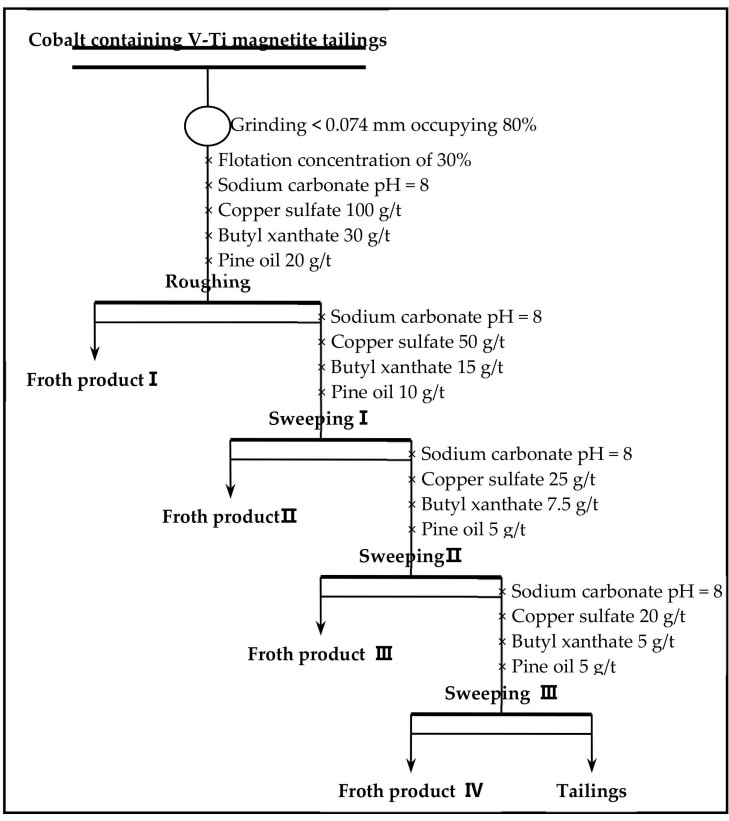

**Figure 8.** Time on flotation sweeping flowsheet.

**Table 3.** Effect of scavenging time on cobalt grade and recovery in flotation (%).

| Products | Yiled | | Cobalt Grade | | Cobalt Recovery | |
|---|---|---|---|---|---|---|
| | Individual | Cumulative | Individual | Cumulative | Individual | Cumulative |
| Froth product I | 1.65 | 1.65 | 1.66 | 1.65 | 86.21 | 86.21 |
| Froth product II | 0.12 | 1.77 | 0.76 | 1.6 | 2.87 | 89.08 |
| Froth product III | 0.08 | 1.85 | 0.53 | 1.55 | 1.33 | 90.41 |
| Froth product IV | 0.04 | 1.89 | 0.26 | 1.53 | 0.33 | 90.74 |
| Tailings | 98.11 | 100.00 | 0.003 | 0.032 | 9.26 | 100.00 |
| Totals | 100.00 | | 0.032 | | 100.00 | |

The flotation process of vanadium–titanium magnetite tailings after one roughing and three sweepings can obtain the separation index of cobalt–sulfur concentrate, with a cobalt grade of 1.53% and cobalt recovery of 90.74%. Compared with the primary roughing, the grade of cobalt decreased by (1.65−1.53)% = 0.12%, and the recovery of cobalt increased by (90.74−86.21)% = 4.53%. The index increased slightly, which was related to the occurrence of cobalt in tailings. Increasing the number of sweeping is mainly reflected in increasing the number of flotation machine slots in production, thus increasing the investment of pre-equipment. However, considering the high economic value of cobalt, although the increase of comprehensive recovery of cobalt by sweeping is relatively small, from the perspective of comprehensive utilization of resources, it is still reasonable to obtain cobalt–sulfur crude concentrate by flotation process of one roughing and three sweeping. In addition, considering the low grade of cobalt–sulfur concentrate, further improvement is also needed.

### 3.1.8. Effect of Time on Flotation Cleaning Process

Through the effect experiment of flotation sweeping on cobalt grade and recovery rate, with the flotation process of one roughing and three sweeping, cobalt–sulfur concentrate with a cobalt grade of 1.53% and cobalt recovery of 90.74% was obtained. According to the quality standard of

cobalt–sulfur concentrate (see Table 4), the particle size of cobalt concentrate is less than 0.175 mm, the moisture content of cobalt concentrate is less than 12%, and inclusions are not allowed in concentrate products. Therefore, according to traditional standards, the cobalt concentrate can be directly used as cobalt concentrate.

**Table 4.** Quality standard of cobalt–sulfur concentrate (%).

| Grades | Composition | | | | | |
| | Co | S | Impurities$\leq$ | | | |
| | $\geq$ | $\geq$ | Cu | Mn | SiO2 | As |
| --- | --- | --- | --- | --- | --- | --- |
| A | 0.50 | 27.00 | 0.40 | 0.03 | 5.00 | 0.04 |
| B | 0.45 | 27.00 | 0.50 | 0.04 | 7.00 | 0.06 |
| C | 0.40 | 27.00 | 0.60 | 0.06 | 10.00 | 0.06 |
| D | 0.35 | 27.00 | 0.70 | 0.08 | 13.00 | 0.08 |
| E | 0.30 | 27.00 | 1.00 | 0.10 | 16.00 | 0.08 |
| F | 0.25 | 27.00 | 1.20 | 0.10 | 18.00 | 0.10 |
| G | 0.20 | 27.00 | 1.20 | 0.10 | 20.00 | 0.10 |

According to the requirements of cobalt concentrate products on the market at present, the quality requirements of cobalt concentrate for existing cobalt production enterprises are as follows: Cobalt grade is generally required to be more than 2% and some enterprises require it more than 6%. Therefore, through the flotation process of one roughing and three sweeping, the cobalt–sulfur concentrate was obtained in this study. The technological process is shown in Figure 9 and the results are shown in Table 5.

**Table 5.** Effect of cleaning time on cobalt grade and recovery in flotation (%).

| Products | Yield | | Cobalt Grade | | Cobalt Recovery | |
| | Individual | Cumulative | Individual | Cumulative | Individual | Cumulative |
| --- | --- | --- | --- | --- | --- | --- |
| Concentrate | 0.67 | 0.67 | 3.33 | 3.33 | 69.71 | 69.91 |
| Middlings VI | 0.12 | 0.79 | 1.65 | 3.07 | 6.19 | 75.9 |
| Middlings III | 0.19 | 0.98 | 0.78 | 2.63 | 4.63 | 80.53 |
| Middlings II | 0.35 | 1.33 | 0.46 | 2.06 | 5.03 | 85.56 |
| Middlings I | 0.58 | 1.91 | 0.29 | 1.52 | 5.25 | 90.81 |
| Tailings | 98.09 | 100.00 | 0.003 | 0.032 | 9.19 | 100.00 |
| Totals | 100.00 | | 0.032 | | 100.00 | |

It is known that the improvement of cobalt grade is relatively small with the increase of concentrating times from Table 5. Through the flotation process of one roughing, three sweeping, and four concentrating, the highest cobalt separation indexes can be obtained from the titanium tailings of vanadium–titanium magnetite separation: Cobalt grade of 3.33% and cobalt recovery of 69.71%. Compared with the indexes of cobalt concentrate before concentration, the cobalt grade increased by (3.33−1.52)% = 1.81% and cobalt recovery rate decreased by (90.81−69.71)% = 11.79%. The separation index of cobalt–sulfur concentrate does not improve much, which further indicates that it is difficult to obtain high-quality cobalt concentrate by a single flotation method. Therefore, it is reasonable to adopt a three-stage concentration process, which could foster the cobalt grade of 2.06% and the cobalt recovery rate of 85.56%.

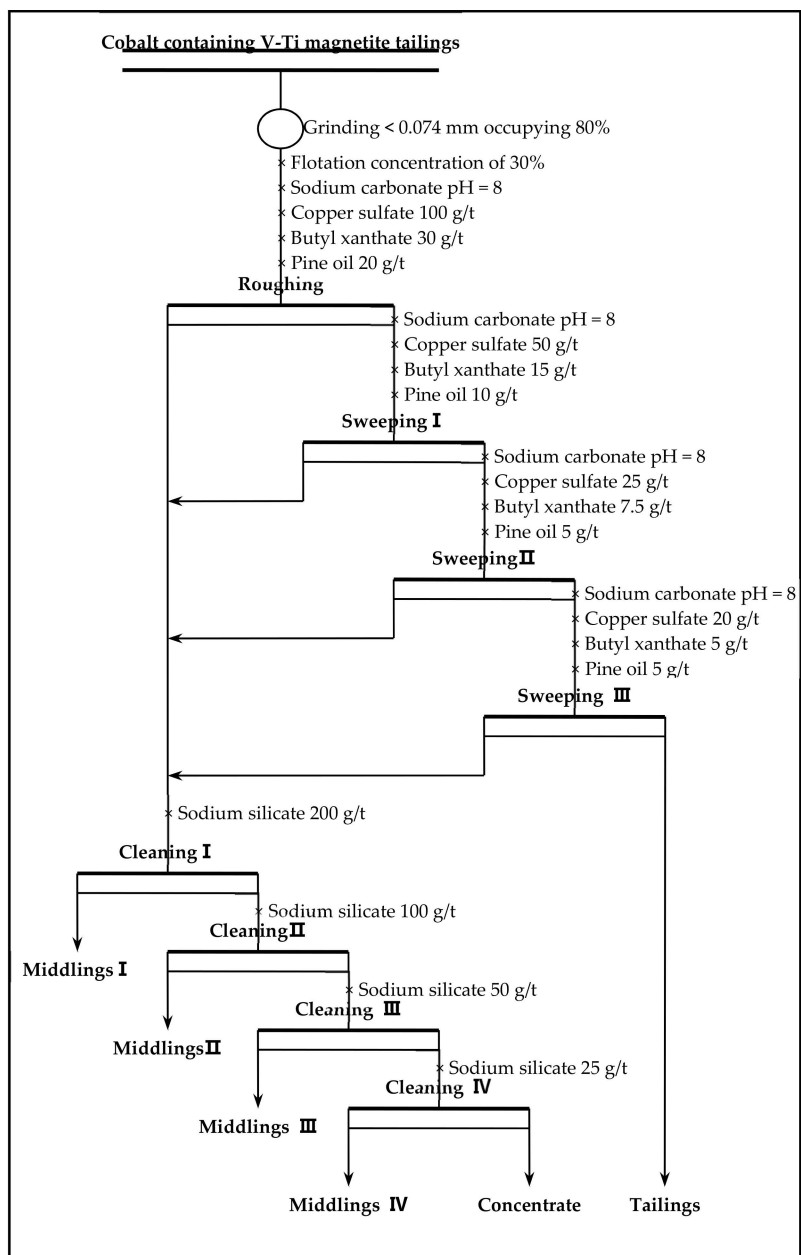

**Figure 9.** Time on flotation cleaning test flowsheet.

### 3.2. The Whole Flotation Flowsheet Test of Recovering Cobalt and Sulfur

The test results of single flotation process conditions on cobalt separation index, sweeping process, and concentrating process show that cobalt is mostly isomorphic in the form of isomorphic substance due to its complex occurrence, fine particle size, and close relationship with sulfide minerals such as pyrite and pyrrhotite. In the flotation process, cobalt is separated from cobalt-bearing materials. Essentially, the flotation of cobalt-bearing minerals can significantly improve the comprehensive recovery of cobalt by roughing and sweeping. The grade of cobalt can be raised to more than 2% by concentrating, but there is still a certain gap from the market requirements for the quality of cobalt concentrate. However, cobalt concentrate can be obtained in advance by flotation, so that cobalt can be pre-enriched to provide important raw materials for subsequent chemical mineral processing, such as roasting and leaching.

Combined with the technological mineralogical characteristics of cobalt, technological conditions test, sweeping, and concentrating test results, the separation process of cobalt is optimized and the

closed-circuit flotation process of one roughing, three sweeping, and three concentrating is adopted. The gangue minerals are further inhibited by increasing the amount of sodium silicate in the concentrating process to ensure that the cobalt grade of the cobalt concentrate is increased to more than 2% and the test indexes of the whole process flow are also examined. The flotation process (shown as Figure 10) was used to carry out the whole process test. The results are shown in Table 6.

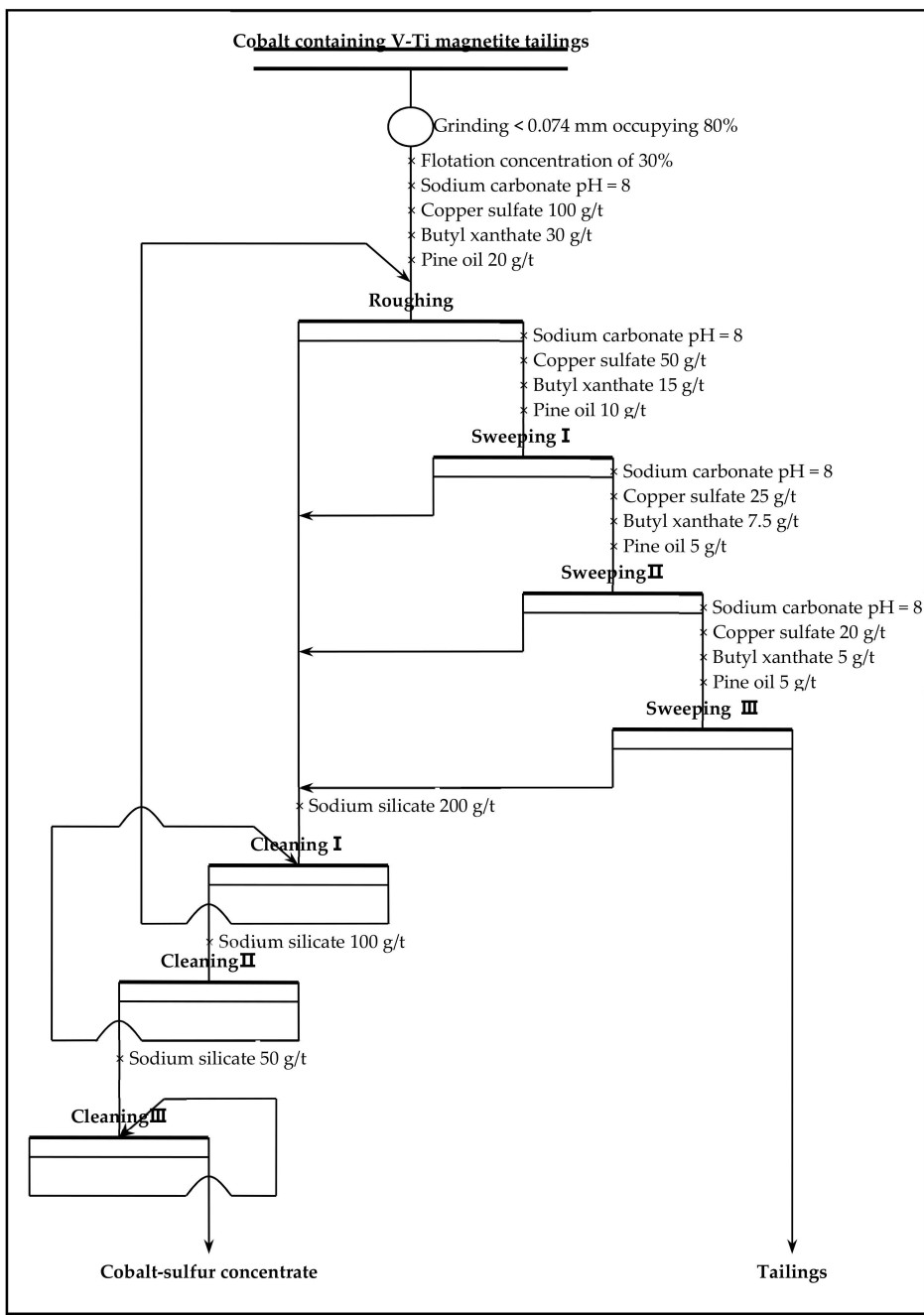

**Figure 10.** The whole flotation test flowsheet of recovering cobalt.

**Table 6.** Results of the whole flotation test flowsheet of recovering cobalt (%).

| Products | Yield | Grade | | Recovery | |
|---|---|---|---|---|---|
| | | Co | S | Co | S |
| Cobalt-sulfur concentrate | 1.32 | 2.08 | 36.12 | 84.77 | 85.79 |
| Tailings | 98.68 | 0.005 | 0.08 | 15.23 | 14.21 |
| Totals | 100.00 | 0.033 | 0.56 | 100.00 | 100.00 |

The Panxi Region is known for low-grade cobalt-bearing vanadium titanomagnetite tailings. This study adopted the flotation process of one roughing, three sweepings, and three cleanings, obtaining the cobalt–sulfur concentrate with a cobalt grade of 2.08%, sulfur content of 36.12%, cobalt recovery rate of 84.77%, and sulfur recovery rate of 85.79, which were superior to the single-condition test results, and further showed that the flotation process had good repeatability.

### 3.3. Cobalt and Sulfur Separation of Cobalt–Sulfur Concentrate

The cobalt–sulfur concentrate with a cobalt grade of 2.08% and sulfur grade of 36.12% was obtained by flotation process of one roughing, three sweepings, and three concentrations. The separation test of cobalt and sulfur was carried out using the technological process of Figure 11 to examine the possibility of separating cobalt and sulfur, respectively, from cobalt concentrate and sulfur concentrate products. The results are shown in Table 7.

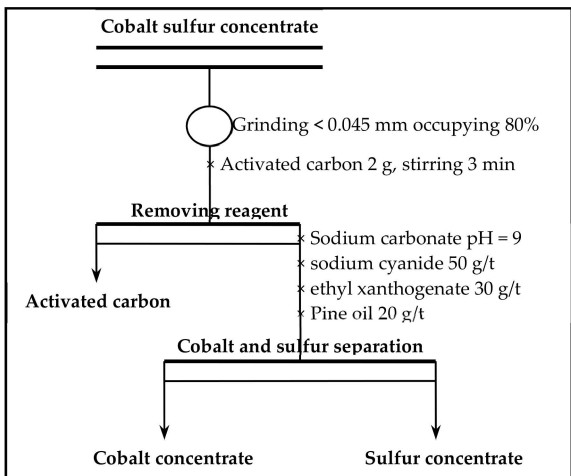

**Figure 11.** Cobalt and sulfur separation test flowsheet of cobalt–sulfur concentrate.

**Table 7.** Results of cobalt and sulfur separation in flotation (%).

| Products | Yield | Grade | | Recovery | |
|---|---|---|---|---|---|
| | | Co | S | Co | S |
| Cobalt concentrate | 49.56 | 2.23 | 38.78 | 53.17 | 53.21 |
| Sulfur concentrate | 50.44 | 1.93 | 33.51 | 46.83 | 46.79 |
| Totals | 100.00 | 2.08 | 36.12 | 100.00 | 100.00 |

There is little difference between cobalt grade of cobalt concentrate and that of sulfur concentrate, and there is a trend whereby the higher the cobalt grade, the higher sulfur grade. According to the mineral composition analysis of cobalt-bearing vanadium titanomagnetite tailings, cobalt-bearing minerals are mainly pyrite and nickel pyrite, pyrite and cobalt sulfide ore, and nickel pyrite in the flotation process. There is little difference in flotability between ores [30–32]. Flotation can be utilized to recover cobalt and sulfur simultaneously, but it is difficult to separate cobalt and sulfur. Therefore,

for the cobalt-bearing vanadium titanomagnetite tailings in the Panxi Region, the cobalt and sulfur concentrate with a cobalt grade of more than 2% can be enriched by flotation. The valuable elements, cobalt and sulfur, in cobalt and sulfur concentrate can also be further separated by roasting and wet leaching.

## 3.4. Discussion

Low grade cobalt-bearing vanadium titanomagnetite tailings from iron and titanium separation in the Panxi Region were treated by a flotation process. The separation indexes of cobalt–sulfur concentrate with a cobalt grade of 2.08%, sulfur content of 36.12%, and cobalt recovery of 84.77% were obtained, realizing the recovery of valuable element cobalt. The results of separation tests of cobalt and sulfur also show that further separation of cobalt and sulfur by the flotation method is difficult, and other chemical beneficiation processes are needed. So, it is necessary to find out the reason for the difficult separation of cobalt–sulfur concentrations by analysis and detection means. The main chemical composition of cobalt–sulfur concentrate was analyzed by XRF smear quantitative analysis method. The results are shown in Table 8. The cobalt–sulfur concentrate was analyzed and characterized by X-ray diffraction (XRD), mineral liberation analyzer (MLA), scanning electron microscope (SEM), and energy disperse spectroscopy (EDS). XRD analysis of cobalt–sulfur concentrate is shown in Figure 12, MLA analysis in Table 9 and Figure 13, SEM analysis in Figure 14, and EDS analysis in Figure 15.

**Table 8.** Main chemical composition analysis of cobalt–sulfur concentrate (%).

| Composition | Fe | Co | TiO$_2$ | S | Cu | SiO$_2$ | CaO | Al$_2$O$_3$ | As | Mn |
|---|---|---|---|---|---|---|---|---|---|---|
| Content | 33.12 | 2.08 | 1.22 | 36.13 | 0.11 | 6.22 | 4.68 | 7.55 | 0.006 | 0.04 |

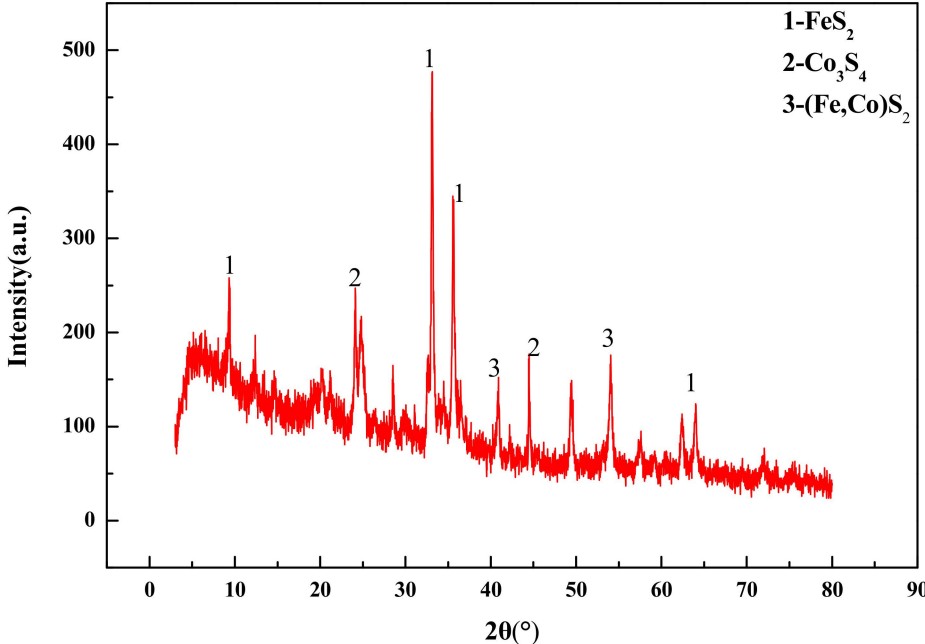

**Figure 12.** XRD analysis of cobalt–sulfur concentrate.

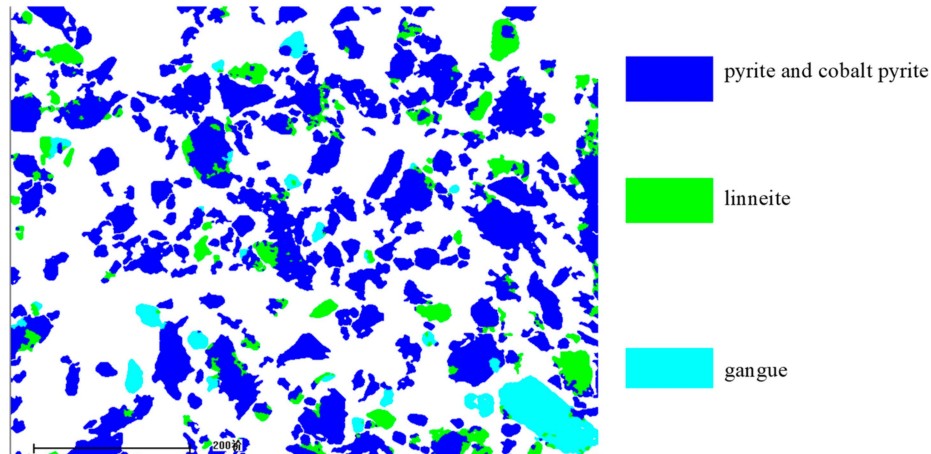

**Figure 13.** Mineral liberation analyzer (MLA) analysis images of cobalt–sulfur concentrate.

**Table 9.** MLA mineral phase composition analysis of cobalt–sulfur concentrate.

| Mineral Phase | Pyrite and Cobalt Pyrite | Linneite | Gangue Mineral | Totals |
|---|---|---|---|---|
| Wt% | 65.64 | 22.64 | 11.72 | 100.00 |

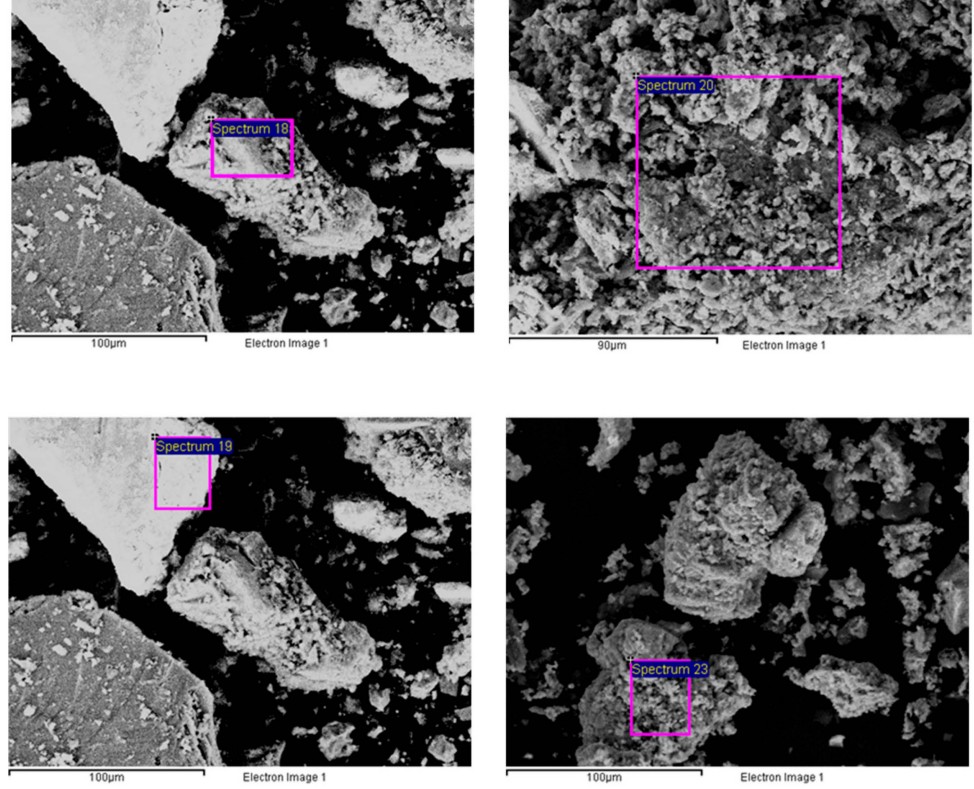

**Figure 14.** SEM analysis images of cobalt–sulfur concentrate.

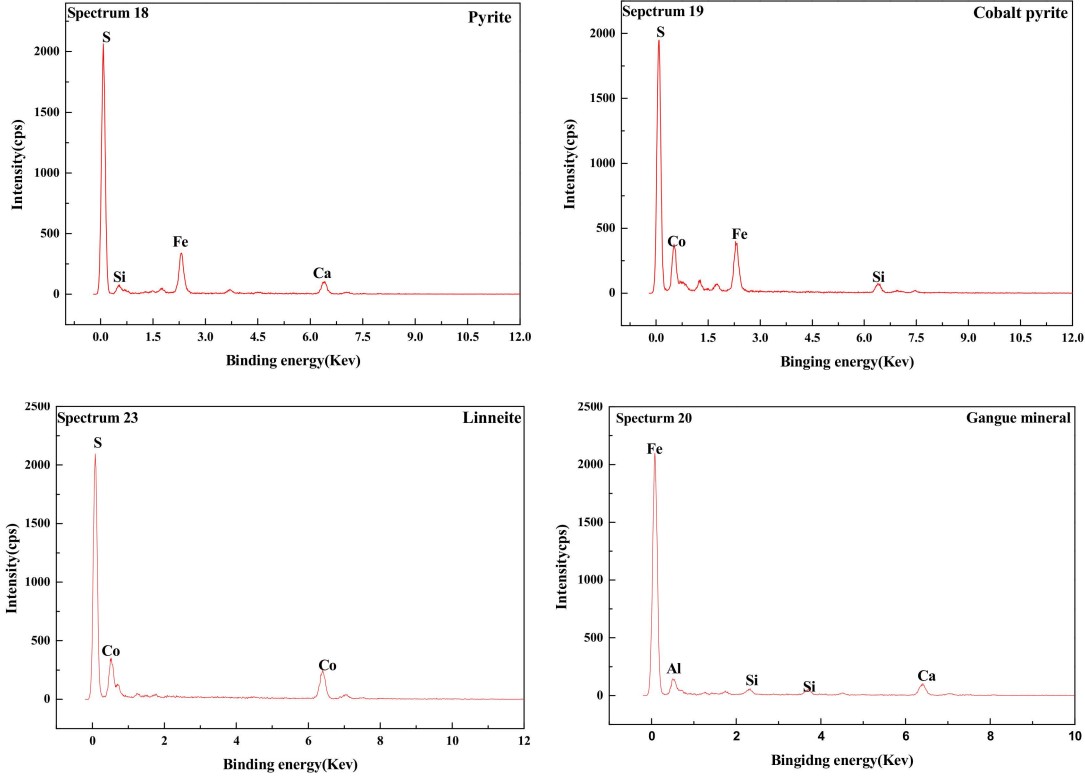

**Figure 15.** Energy disperse spectroscopy (EDS) spectrum images analysis of cobalt–sulfur concentrate.

It is known that the main minerals in cobalt–sulfur concentrate are $FeS_2$, $Co_3S_4$, and $(Fe,Co)S_2$, with $FeS_2$ and $(Fe,Co)S_2$ accounting for 65.64%, $Co_3S_4$ for 22.64%, and gangue minerals for 11.72%. Cobalt in cobalt pyrite is closely related to pyrite in the form of isomorphism, and the flotability difference between $(Fe,Co)S_2$ and $FeS_2$ is smaller and it is very difficult to select reasonable inhibitors to achieve separation, which makes it difficult to separate cobalt and sulfur by flotation. Therefore, cobalt preconcentration is easy realized in the actual flotation separation process [33–35]. Cobalt–sulfur concentrate can be used as raw material for further separation of cobalt and sulfur in smelting by pyrometallurgical or hydrometallurgical methods.

## 4. Conclusions

(1) The low grade cobalt-bearing vanadium titanomagnetite tailings in the Panxi Region contained a cobalt grade of 0.032% and sulfur grade of 0.56%. The main recovered valuable elements are cobalt and sulfur. The main metal sulfide minerals in the tailings are $FeS_2$, $Fe1-xS$, $Co_3S_4$, and $(Fe,Co)S_2$.

(2) Cobalt and sulfur were recovered in low-grade cobalt-bearing V–Ti tailings by a flotation process of one roughing, three sweepings, and three cleanings. This study obtained the separation indexes of cobalt–sulfur concentrate, with a cobalt grade of 2.08%, sulfur content of 36.12%, cobalt recovery of 84.77%, and sulfur recovery of 85.79%.

(3) Studies of the characterization and analysis of cobalt–sulfur concentrate by XRD, MLA, SEM, and EDS indicated that the main minerals in cobalt–sulfur concentrate are $FeS_2$, $Fe1-xS$, $Co_3S_4$, and $(Fe,Co)S_2$, of which $FeS_2$ and $(Fe,Co)S_2$ account for 65.64%, $Co_3S_4$ for 22.64%, and gangue mineral for 11.72%. Cobalt in cobalt pyrite is closely related to pyrite in the form of isomorphism, and the flotability difference between cobalt and pyrite is small, which makes it difficult to further separate cobalt and sulfur by flotation. Preconcentration of cobalt and sulfur was appropriate from low-grade cobalt-bearing V–Ti tailings. Cobalt–sulfur concentrate can be processed using pyrometallurgical or hydrometallurgical methods, such as oxidation roasting and pressure leaching, and independent cobalt

and sulfur products can be obtained from cobalt—sulfur concentrate. The efficient utilization of cobalt and sulfur of low-grade cobalt-bearing V–Ti tailings in the Panxi Region can be realized finally.

**Author Contributions:** This is a joint work of the two authors; each author was in charge of their expertise and capability: J.X. for conceptualization, methodology, validation, original draft preparation, and writing, Y.Z. for formal analysis and investigation

**Funding:** This work was supported by the Sichuan Science and Technology Program (Grant Nos. 2019YFS0451, 2018FZ0092 and 2019YFS0452); Key Laboratory of Sichuan Province for Comprehensive Utilization of Vanadium and Titanium Resources Foundation(2018FTSZ35); China Geological Big Survey (Grant No. DD20190694); Open Foundation of the State Key Laboratory of Refractories and Metallurgy, Wuhan University of Science and Technology (Grant No. ZR201801); Open Foundation of the Key Laboratory of Radioactive and Rare and Sparse Minerals of the Ministry of Land and Resources (Grant No. RRSM-KF2018-02).

**Conflicts of Interest:** The authors declare no conflict of interest. The funders had no role in the design, analyses, and interpretation of any data of the study.

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
