# Peer review of "Recovering Cobalt and Sulfur in Low Grade Cobalt-Bearing V–Ti Magnetite Tailings Using Flotation Process"

_processes, doi:10.3390/pr7080536_

Round 1

Reviewer 1 Report

I suggest unifying the naming of flotation reagents namely collectors and the frother used in this study. I saw somewhere in the text  the naming of xanthine and xanthate. Personally, I suggest using xanthate everywhere. The same situation concerning the frother which named sometimes pine oil or terpineol.

Moreover, I suggest to using concerning the stages of flotation Rouhging, scavenging and cleaning instead of using sweeping in place of scavenging.

Besides, I suggest the author to discuss merely the obtained results instead of giving first all the theorical basis of the studied parameter before presenting or selecting the best value. The aforementionned theorical basis can be used to support the arguments while citing some relevant papers with the aim of justifying their statements. The way the authors discuss the obtained resultats needs to be revised.

The paper is of great value because it is dealing with an interesting topic. However, I am not sure that the processing suggested by authors for the separation of cobalt from sulphur is environmentally acceptable: the roasting of concentrates followed by the metal leaching may for its recovery as a precipitate or as cathode. The authors needs to put a stress on environmental issues to the process implementation such as the recovery of SO2 and its possible conversion into H2SO4.

Author Response

Response to Reviewer 1 Comments

Point 1: I suggest unifying the naming of flotation reagents namely collectors and the frother used in this study. I saw somewhere in the text the naming of xanthine and xanthate. Personally, I suggest using xanthate everywhere. The same situation concerning the frother which named sometimes pine oil or terpineol.

Response 1: Thanks for your precious suggestions. All of them have been revised in the article, please review it.

Point 2: Moreover, I suggest to using concerning the stages of flotation Rouhging, scavenging and cleaning instead of using sweeping in place of scavenging.

Response 2: Thank you very much for your suggestion again. It has been revised in the article after checking.

Point 3: Besides, I suggest the author to discuss merely the obtained results instead of giving first all the theorical basis of the studied parameter before presenting or selecting the best value.

Response 3: According to your precious review opinions, the condition test have been adjusted in the article.

Point 4: The aforementionned theorical basis can be used to support the arguments while citing some relevant papers with the aim of justifying their statements. The way the authors discuss the obtained resultats needs to be revised.

Response 4: According to your precious review opinions, the basic theoretical research of relevant literature was consulted, and the results of the paper were revised.

Point 5: The paper is of great value because it is dealing with an interesting topic. However, I am not sure that the processing suggested by authors for the separation of cobalt from sulphur is environmentally acceptable: the roasting of concentrates followed by the metal leaching may for its recovery as a precipitate or as cathode. The authors needs to put a stress on environmental issues to the process implementation such as the recovery of SO2 and its possible conversion into H2SO4.

Response 5: Thanks for the recognition of our research work. The main mineral in cobalt sulfur concentrate is still pyrite. Pyrite is one of the raw materials for preparing sulfuric acid. In industry, SO2 and iron oxide slag (Fe3O4 or Fe2O3) are obtained by using mature and reliable oxidation roasting process. In this study, cobalt sulfur concentrate obtained from low grade cobalt-bearing vanadium titanomagnetite can also be further treated by oxidation roasting process, and cobalt products can be further prepared by ammonia leaching process. At present, we are carrying out the experimental study on the oxidation roasting and combined ammonia salt leaching technology of cobalt sulfur concentrate, and the cobalt leaching index with the cobalt leaching rate higher than 95% can be obtained, and the cobalt separation index is relatively ideal.

Finally, we sincerely hope that you can see our efforts to the revised work, and also hope to get your approval.

Reviewer 2 Report

Congratulations on the research idea and correctness of the measurement method.

Author Response

Response to Reviewer 2 Comments

Point 1: Congratulations on the research idea and correctness of the measurement method.

Response 1: Thank you very much for your recognition of our study work and your hard review. At present, we are carrying out the separation test of cobalt-sulfur concentrate, and we hope that we can make valuable Suggestions for our research work next time.

Reviewer 3 Report

presented is work into the use of multiple flotation flowsheeting to recover cobalt from low grade tailings.  The authors certainly present a large number of experiments to optimise the system.  However, i feel the discussion is largely very cursory and qualitative in nature. At times, it felt more like a technical report rather than a research paper. The authors explain what was found by varying a variety of parameters, but not the underlying mechanisms as to why observed flotation behaviour was found.  There was also almost a complete lack of comprehensive linking to previous literature for justification within their discussion as well. Hence, it was very difficult as a reader to understand whether the results were largely expected or not, or where the real novelty and importance lay.  Part of the problem was there was really too many preliminary tests shown in the manuscript, some of which would have been much more suitable to be placed in a supplementary materials, to better focus on the importance science. 

This lack of focus on the real novelty of the work was also shown in the introduction. While the problem was clearly presented, the introduction was too brief on covering the current state-of-the-art, and how this work builds on previous research. 

Overall, i cannot recommendation acceptance of the paper in its current form. 

Author Response

Response to Reviewer 3 Comments

Point 1: presented is work into the use of multiple flotation flowsheeting to recover cobalt from low grade tailings. The authors certainly present a large number of experiments to optimise the system. However, I feel the discussion is largely very cursory and qualitative in nature. At times, it felt more like a technical report rather than a research paper.

Response 1: Thanks for your precious suggestions. We have reviewed a large amount of literature and reorganized the discussion of the results into a language description. I hope you can see our revision work.

Point 2: The authors explain what was found by varying a variety of parameters, but not the underlying mechanisms as to why observed flotation behaviour was found.

Response 2: This paper lay particular stress on technique research, this study through a lot of test research were carried out to find the effects on cobalt grade and recovery, the ideal cobalt-sulfur concentrate index were obtained. The technological mineralogy for the sulphur concentrate cobalt analysis, for the flotation separation of cobalt, sulfur, provides a important basis and research, as well as sulphur concentrate cobalt further separation of cobalt sulphur provides an important guiding role.

Point 3: There was also almost a complete lack of comprehensive linking to previous literature for justification within their discussion as well. Hence, it was very difficult as a reader to understand whether the results were largely expected or not, or where the real novelty and importance lay.

Response 3: According to your valuable suggestions, we have consulted relevant literature and improved the leading part of the article. Please review it.

Point 4: Part of the problem was there were really too many preliminary tests shown in the manuscript, some of which would have been much more suitable to be placed in a supplementary materials, to better focus on the importance science.

Response 4: Thank you for your suggestion. Since the low-grade cobalt-vanadium titanomagnetic tailings in Panxia region are secondary resources and the content of valuable cobalt is very low, the starting point of this study is to try to adopt a simple process to treat the tailings and realize the effective recovery of valuable element cobalt and sulfur.

Point 5: This lack of focus on the real novelty of the work was also shown in the introduction. While the problem was clearly presented, the introduction was too brief on covering the current state-of-the-art, and how this work builds on previous research. Overall, I cannot recommendation acceptance of the paper in its current form.

Response 5: Thank you very much for your great comments. This paper focuses on technological research and provides some guidance for the comprehensive utilization of low-grade cobalt-bearing vanadium titanium magnetite in Panxi region through experimental research. Meanwhile, we are conducting the separation test of cobalt-sulfur concentrate, and the separation effect of cobalt and sulfur is obvious. We have made a large number of revisions to this paper, hoping to gain your approval. And if you have any questions, please contact us. We look forward to hearing from you.

Round 2

Reviewer 3 Report

the authors have significantly improved their manuscript and i am now happy to recommend its publication. I still feel some of the large number of process schematics could be put in a supplementary information to reduce the length of the main article. However, that is authors choice